# The Association between Electronic Health Literacy and Oral Health Outcomes among Dental Patients in Saudi Arabia: A Cross-Sectional Study

**DOI:** 10.3390/healthcare11121804

**Published:** 2023-06-20

**Authors:** Faisal F. Hakeem, Ismail Abdouh, Hatem Hazzaa Hamadallah, Yunus Osama Alarabi, Abdulrahman Saad Almuzaini, Majed Maher Abdullah, Ammar Abdulrahman Altarjami

**Affiliations:** 1Department of Preventive Dental Sciences, College of Dentistry, Taibah University, Al-Madinah Al-Munawwarah 42353, Saudi Arabia; 2Centre for Epidemiology Versus Arthritis, The University of Manchester, Manchester M13 9PL, UK; 3Department of Oral Basic and Clinical Sciences, College of Dentistry, Taibah University, Al-Madinah Al-Munawwarah 42313, Saudi Arabia; iabdouh@taibahu.edu.sa; 4College of Dentistryand Hospital, Taibah University, Al-Madinah Al-Munawwarah 42353, Saudi Arabia; hatem.xx@taibahu.edu.sa (H.H.H.); yunus1999@taibahu.edu.sa (Y.O.A.); tu3902398@taibahu.edu.sa (A.S.A.); majed2bdullah@taibahu.edu.sa (M.M.A.); tu3901870@taibahu.edu.sa (A.A.A.)

**Keywords:** oral health, eHealth literacy, health literacy, health behavior, tooth loss, brushing frequency, dental patients

## Abstract

Objective: This cross-sectional study aimed to investigate the association between electronic health (eHealth) literacy and oral health outcomes, including the number of teeth and brushing frequency. Methods: A total of 478 participants were included in the study and assessed for their eHealth literacy levels. Demographic variables, including age, gender, income, and education, were collected. The participants’ number of teeth and brushing frequency were also recorded. Multiple regression analyses were performed to examine the relationship between eHealth literacy and oral health outcomes, adjusting for sociodemographic variables. Results: The study sample consisted of both males (66.5%) and females (33.5%), with a mean age of 31.95 years. Among the participants, 16.95% were classified as having inadequate eHealth literacy, 24.06% had problematic eHealth literacy, and the majority (59.00%) demonstrated sufficient eHealth literacy. There was a significant association between eHealth literacy and oral health outcomes. Individuals with problematic eHealth literacy had a higher likelihood of having a greater number of teeth (RR = 1.12, 95% CI: 1.05–1.20, *p* < 0.001) compared to those with inadequate eHealth literacy. Similarly, individuals with sufficient eHealth literacy showed a higher likelihood of having more teeth (RR = 1.14, 95% CI: 1.07–1.21, *p* < 0.001) compared to the inadequate eHealth literacy group controlling for age, gender, income, and education. Individuals with problematic eHealth literacy exhibited a tendency towards lower odds of irregular brushing (OR = 0.39, 95% CI: 0.15–1.02, *p* = 0.054), although this result was marginally significant. In contrast, individuals with sufficient eHealth literacy had significantly lower odds of irregular brushing frequency (OR = 0.24, 95% CI: 0.10–0.62, *p* = 0.003) compared to the inadequate eHealth literacy group. Conclusion: The findings suggest a positive association between eHealth literacy and oral health outcomes. Improving eHealth literacy may have implications for promoting better oral health behaviors and outcomes.

## 1. Introduction

As the focus of healthcare has shifted from treating illnesses to preserving health, health promotion has emerged as a societal issue that affects individuals in general as well as healthcare professionals [1]. During the past decades, there has been an increase in the use of the Internet and digital devices, which has led to the integration of health within these aspects [2]. Electronic health (eHealth) has been defined as healthcare services and health information provided and obtained using electronic and digital means [3]. A conceptual model of eHealth has adequately integrated technology, business, and health to create a new idea of service delivery in the healthcare setting [4]. eHealth benefits include increasing the effectiveness and accessibility of medical and dental services; for example, the implementation of various mobile health applications and social media. Furthermore, eHealth offers several advantages, including the improved efficiency and accessibility of medical services. With the help of the Internet, people who face difficulties in accessing medical services can now obtain health information and receive assistance in managing their oral health. This has made healthcare more convenient and accessible, especially for those living in remote areas or with mobility impairments. Additionally, the use of eHealth technologies has helped to reduce waiting times, increase patient engagement, and improve overall healthcare outcomes [5]. Today, the Internet is considered a major source of health information. People who have difficulty obtaining medical and dental care can access health information online and receive assistance in managing their health conditions. However, obtaining health-related information from the internet requires certain skills related to health literacy, such as reading, understanding, and appraising information [6]. Thus, the concept of eHealth Literacy emerged and was defined as “the ability to seek, find, understand, and appraise health information from electronic sources and apply the knowledge gained to addressing or solving a health problem” [7]. When individuals have low eHealth literacy, they may face challenges in effectively using health information from the internet. They may encounter difficulties in finding the reliable information they need, or they may come across distorted or inaccurate information that can lead to health imbalances and disparities. Therefore, it is essential to improve the level of eHealth literacy among individuals, enabling them to discern and use accurate information [8].

In the field of dentistry and oral health, the concept of health literacy has gained importance and has been integrated into oral health literacy (OHL), which encompasses various tools and resources [9,10]. Research has shown that OHL is influenced by both individual and systemic factors. Individual factors, such as educational level, language proficiency, and health literacy, can impact a person’s ability to comprehend and utilize oral health information. Similarly, systemic factors, including the complexity and demands of the healthcare system, can also affect OHL [11,12,13,14]. When oral health information is presented in a complex or confusing manner, patients may struggle to understand and apply it, potentially leading to unfavorable oral health outcomes [15]. Having a solid foundation of oral health knowledge equips individuals to practice proper self-oral care, handle potential emergencies at home, and make informed decisions about dental treatments. As individuals acquire more knowledge about oral health, their attitudes toward oral care may change, resulting in the adoption of healthier oral health behaviors. This, in turn, can positively impact overall oral health by preventing common issues such as gingival disease and tooth decay [16]. By establishing a health-literate system of care, healthcare providers can enhance the accessibility and comprehensibility of oral health information for patients. This can be achieved through simplified communication, the use of plain language, and the provision of visual aids to help patients understand oral health information [17,18].

Individuals who possess greater expertise and competence in accessing eHealth information tend to be more effective at managing their own health and adopting healthier behaviors. This suggests that the ability to access and use eHealth resources, such as online health portals, mobile health applications, and other digital tools, can facilitate a more informed and engaged approach to healthcare [19]. Through the utilization of these technologies, individuals can gather information on their medical conditions, track their symptoms and progress, communicate with healthcare providers, and make more informed decisions regarding their health [20]. Indeed, the term teledentistry is considered an example of such a concept that refers to the provision of dental care, advice, or treatment using online communication technology, such as video conferencing, remote monitoring, and online consultations [21]. This innovation has revolutionized the dental industry, improving access to oral care, and promoting better oral health behavior. In fact, teledentistry plays a crucial role in promoting oral health literacy; as patients engage in virtual consultations and access oral health information online, they become adept at navigating digital health resources and making informed decisions about their oral care [21].

Over the past few decades, Saudi Arabia has undergone rapid socioeconomic development and related lifestyle modifications, but the prevalence of poor oral health in Saudi Arabia is still rising [22]. Understanding the level of eHealth literacy among the population and addressing the disparities in eHealth literacy can improve health outcomes and reduce healthcare costs. Healthcare providers, policymakers, and health educators should work together to develop and implement eHealth education programs targeting disadvantaged groups with low eHealth literacy. As far as we know, no previous studies have investigated the association between electronic health literacy and oral health among the Saudi population at the time of conducting this study. Therefore, the purpose of this study was to determine if eHealth literacy is associated with oral health status and oral health behaviors among a sample of patients who visited the Taibah University College of Dentistry Hospital.

## 2. Materials and Methods

Study Design and Sample: This was a cross-sectional study that was conducted at the Taibah University College of Dentistry Hospital in Madinah, Saudi Arabia, in 2023. The patients included in the study were Arabic-speaking individuals aged 18 and older who sought dental treatment at the hospital and voluntarily agreed to participate. Potential participants were approached during their scheduled appointments to assess their interest and eligibility for participation. Convenience sampling was employed, where patients were selected based on their availability and willingness to participate. We adhered to the STROBE (Strengthening the Reporting of Observational studies in Epidemiology) guidelines in preparing and reporting the findings of this manuscript [23].

Ethical Considerations: The study was approved by the Institutional Review Board of Taibah University (TUCDREC/O10323) and followed the guidelines of the Declaration of Helsinki. Each participant signed a consent form. Full confidentiality of the collected information was provided to the research participants.

eHealth Literacy: eHealth literacy was the main explanatory variable of this study. The validated Arabic version of the Electronic Health Literacy Scale (eHEALS) questionnaire was used to assess participants’ perceived ability to find, evaluate, and use electronic health information [24]. The questionnaire provided a total score ranging from 8 to 40, indicating the level of eHealth literacy. The eHEALS scores were categorized into three threshold values: inadequate (8–20 points), problematic (21–26 points), and sufficient (27–40 points) following previous validation studies [25].

Outcome Variables: Oral health was the main outcome of this study. We had two oral health outcomes: the number of teeth and brushing frequency. The number of teeth was assessed through a clinical examination using the WHO method [26]. A trained and calibrated dental examiner (dental intern at the College of Dentistry, Taibah University) examined participants on a dental chair using a mouth mirror and a periodontal probe. The examiner’s calibration process, following the WHO Oral Health Methods, involved two stages, ultimately yielding an excellent inter-rater reliability statistic (Kappa) of 0.98 [26,27]. Remaining roots, implants, and dental prostheses were not included. Thus, the number of teeth for each participant was recorded as the sum of all natural teeth present (0–32). The brushing frequency was assessed using a self-administered questionnaire that asked participants how many times they brushed their teeth per day, with response options ranging from less than once a day to more than three times a day. Participants were grouped into (do not brush, brush at least once/day, and brush twice or more/day) [26].

Covariates: The covariates of the study were the sociodemographic variables that were included in the analysis to adjust for potential confounding factors, these included age, gender, education level, and income. Sociodemographic data were collected through a self-administered questionnaire based on the Saudi General Authority for Statistics classification system. Income was measured by asking participants about their monthly income in Saudi Riyals. The income variable was categorized into four groups: “less than 2500SR”, “2500-4999SR”, “5000-9999SR”, and “More than 10,000SR. The education variable reflects the educational attainment of the study participants. It was divided into four categories: “No education”, “less than 6 years”, “7–12 years”, and “University or higher”.

Data Analysis: First, the variables of the study were summarized across the eHealth literacy groups (inadequate, problematic, and sufficient) using means and standard deviations for continuous variables and frequencies and percentages for categorical variables. The relationship between eHealth literacy and oral health outcomes (number of teeth and brushing frequency) was then examined using negative binomial regression and logistic regression analysis, respectively. in the logistic regression, the brushing frequency was dichotomized into an “irregular brushing” category, which refers to individuals who do not engage in daily tooth brushing, and a “regular brushing” category, which includes individuals who brush their teeth at least once a day, or twice or more a day. In addition, the demographic variables (age, gender, education level, and income) were included as covariates in these analyses to adjust for potential confounding. The magnitude of the association between eHealth literacy and oral health was presented as adjusted incidence rate ratios with corresponding 95% confidence intervals for the negative binomial regression analysis, and an adjusted odds ratio with corresponding 95% confidence intervals for the logistic regression analysis. The statistical analysis was performed using Stata version 17 software, which was developed and distributed by StataCorp LLC (College Station, TX, USA).

## 3. Results

The sample consisted of 478 participants who were assessed for their eHealth literacy levels. In general, the study sample consisted of both males and females, with males accounting for 66.5% of the participants and females accounting for 33.5%. The participants’ mean age was 31.95 years (SD = 12.1). The age range of the participants varied from 18 years to 71 years. In terms of education, the sample included individuals with varying levels of educational attainment. The majority of participants had completed either 7–12 years of education (38.7%) or university or higher education (55.0%). Regarding income, the majority of participants fell into the “less than 2500SR” category, accounting for 53.1% of the sample. In terms of oral health, the mean number of teeth was 26.45, and regarding brushing frequency, the sample was divided into three categories: individuals who did not brush their teeth (9.6%), those who brushed once a day (46.7%), and those who brushed twice a day (43.7%). Among the participants, 16.95% were classified as having inadequate eHealth literacy, 24.06% had problematic eHealth literacy, and the majority (59.00%) demonstrated sufficient eHealth literacy. When comparing the eHealth literacy groups, there were significant differences across the study variables. The inadequate eHealth literacy group was older and exhibited lower levels of education, lower income, a higher proportion of males, fewer teeth, and lower brushing frequency compared to problematic and sufficient eHealth literacy groups [Table 1].

In the regression analysis, after adjusting for age, gender, income, and education, individuals with problematic eHealth literacy had a higher rate ratio (RR) of 1.12 (95% CI: 1.05–1.20, *p* < 0.001), indicating a greater likelihood of having a higher number of teeth compared to those with inadequate eHealth literacy. Similarly, individuals with sufficient eHealth literacy exhibited a higher IRR of 1.14 (95% CI: 1.07–1.21, *p* < 0.001) compared to the inadequate eHealth literacy group after adjusting for sociodemographic variables, suggesting a positive association between eHealth literacy and a greater number of teeth [Table 2, Figure 1]. Regarding brushing behavior, individuals with problematic eHealth literacy exhibited an OR of 0.39 (95% CI: 0.15–1.02, *p* = 0.054), indicating a tendency towards lower odds of irregular brushing compared to those with inadequate eHealth literacy, although this result was marginally significant. Similarly, individuals with sufficient eHealth literacy had significantly lower odds of irregular brushing with an OR of 0.24 (95% CI: 0.10–0.62, *p* = 0.003) compared to the inadequate eHealth literacy group after adjusting for sociodemographic variables [Table 3, Figure 1].

## 4. Discussion

In this study, the aim was to examine the relationship between eHealth literacy and oral health status, as well as oral health behavior, among patients who visited the Taibah University College of Dentistry Hospital. The main findings of the study indicate that participants with better eHealth literacy, classified as problematic or sufficient eHealth literacy, had a significantly higher mean number of teeth compared to those with inadequate eHealth literacy. This suggests that individuals with better eHealth literacy tend to have better oral health status. Furthermore, the study found that participants with better eHealth literacy had a significantly lower odds ratio for not brushing their teeth compared to those with inadequate eHealth literacy. This highlights the importance of eHealth literacy in promoting positive oral health behaviors, such as regular brushing.

To the best of our knowledge, this is the first study to assess the relationship between eHealth literacy and oral health outcomes. However, we found many studies that align with our study in assessing the relationship between health literacy/oral health literacy and oral health outcomes. Some of these studies suggested that individuals with lower health literacy and oral health literacy had less optimum oral health behaviors and a worse oral health status. When compared to individuals with greater literacy, those with poorer health literacy and oral health literacy brush their teeth less frequently compared to those with higher literacy [10,28,29,30,31,32]. They were also shown to have more tooth loss over time [30,31,32], worse periodontal disease status [33], irregular flossing [10], fewer dental visits, and more emergency room visits for nontraumatic dental conditions [14]. These findings suggest that individuals with low health literacy and oral health literacy may face challenges in practicing effective oral hygiene and achieving excellent oral health. Overall, existing evidence indicates a relationship between literacy levels, oral health behaviors, and oral health outcomes, implying that improving health literacy might help minimize oral health inequities and lead to improving dental health on a larger scale. On the other hand, other studies have shown that there is no significant difference in health outcomes and behaviors between individuals with high health literacy and those with low health literacy [34,35]. Moreover, a recent systematic review found inconclusive evidence regarding the association between oral health literacy and oral health outcomes [18]. The inconsistency between the results of different studies on health literacy and oral health literacy outcomes can be due to various factors, including the use of different tools to measure the outcomes and levels of health literacy; the sample size and the type of sampling used in the study also play a significant role. Therefore, studies that have used different tools to measure health literacy produced different results, which can make it challenging to compare the findings. Similarly, the use of different tools to measure oral health outcomes, such as dental caries, periodontal disease, and oral health-related quality of life, can also lead to inconsistent results.

The relationship between eHealth literacy and oral health outcomes could be influenced by several factors and mechanisms. eHealth literacy refers to an individual’s ability to find, understand, and evaluate online health information for informed decision-making [25]. In terms of oral health, eHealth literacy could play a vital role in accessing reliable information, engaging in preventive behaviors, and seeking appropriate dental care. Currently, many countries are undergoing a digital transformation, particularly in accessing dental services through the Internet and digital applications. This shift extends to accessing preventive services, online information, scheduling automatic dental appointments, and receiving post-treatment instructions and medications [36]. The widespread adoption of digital platforms in the dental sector highlights the importance of understanding and promoting eHealth literacy to effectively navigate and benefit from these digital advancements in oral healthcare. This association can be explained through factors such as access to oral health information, understanding and comprehension of online resources [25], evaluation of information quality [37], empowerment and self-efficacy [38], and improved communication with healthcare providers [39]. Higher eHealth literacy enables individuals to access a variety of online sources, comprehend oral health concepts, evaluate information credibility, feel empowered in managing their oral health, and actively participate in shared decision-making. However, it is important to consider potential explanatory variables, such as socioeconomic status, education, digital access, and cultural factors, which may affect eHealth literacy [40]. Addressing disparities in eHealth literacy can promote equitable access to oral health information, enhance skills, and ultimately improve oral health outcomes for individuals and communities.

Socioeconomic status (SES) plays a crucial role in determining health outcomes, including oral health [41]. Higher SES is often associated with better access to healthcare, education, resources, and opportunities, leading to improved health outcomes [42]. In this context, eHealth literacy could serve as an indicator of higher SES and its connection to oral health outcomes [43]. Higher SES individuals tend to have greater access to technology and digital resources, higher educational attainment, proactive health-seeking behaviors, and comprehensive health knowledge. These factors enable them to effectively utilize eHealth resources, engage in informed decision-making, and actively manage their oral health. However, it is important to acknowledge existing disparities, as individuals from lower SES backgrounds may face limitations in terms of technology access and digital literacy skills, which can hinder their eHealth literacy and oral health outcomes [43]. A previous study found that the association between oral health literacy and oral health-related behaviors was no longer significant after adjustment for social class [10]. Addressing these disparities requires targeted efforts to enhance digital access and literacy among individuals from all socioeconomic backgrounds, promoting health equity and improving overall oral health outcomes.

The results of this study have important public health implications. Firstly, they highlight the significance of eHealth literacy in relation to oral health outcomes. Improving eHealth literacy among the population can empower individuals to access, understand, and effectively utilize online health information, leading to better oral health management and behaviors [44]. Secondly, the findings emphasize the need for targeted interventions and educational programs to enhance eHealth literacy, particularly among individuals with low eHealth literacy [45]. Such interventions can focus on improving digital health literacy skills, promoting access to reliable and accurate oral health information online, and enhancing communication between healthcare providers and patients in the digital realm. By addressing these public health implications, policymakers and healthcare professionals can contribute to improving oral health outcomes and promoting health equity in the population.

The integration of eHealth literacy assessment into routine oral health screenings and the implementation of personalized interventions in clinical practice have important implications for improving oral health outcomes. By assessing patients’ eHealth literacy levels, healthcare providers can identify individuals who may need additional support in accessing and utilizing online health information [25,46]. Assessing eHealth literacy in the context of oral health is crucial, especially considering the increasing availability of online oral health services, such as booking appointments for preventive and therapeutic dental care. The rise of telehealth, personal health records, and teledentistry further emphasizes the need to understand individuals’ eHealth literacy skills to effectively navigate and utilize these digital platforms for oral health purposes [47,48]. Tailoring interventions based on patients’ literacy levels enables healthcare providers to address specific barriers to oral health knowledge and behavior change [49], reinforcing existing knowledge for those with sufficient eHealth literacy and enhancing digital health literacy skills for those with inadequate eHealth literacy [50]. This personalized approach also fosters effective communication between providers and patients, allowing for optimal comprehension and engagement. Leveraging digital platforms and technologies further enhances patient education and engagement, providing reliable oral health information, personalized advice, and convenient access to appointments and reminders. Integrating eHealth literacy assessment and digital interventions within clinical practice could potentially strengthen the patient-provider relationship, empower individuals to actively participate in their oral health, and contribute to improved oral health outcomes and reduce disparities [39,51]. Some of the policy recommendations that could be concluded from the study findings include integrating eHealth literacy promotion into public health initiatives [52], establishing guidelines for online oral health information [52], incorporating eHealth literacy training in oral health workforce education [52,53], fostering collaboration between oral health and digital health sectors [54,55], and supporting research on eHealth literacy interventions [56,57]. By prioritizing eHealth literacy, policymakers can empower individuals to access reliable online resources, make informed decisions, and engage in behaviors that promote optimal oral health. These policy recommendations provide a framework for creating supportive environments that facilitate digital health literacy, ultimately enabling individuals to take an active role in managing their oral health and contributing to improved oral health outcomes. Considering that our study focused on eHealth and oral health in a single country, Saudi Arabia, there are several key directions for further research with the support of the government. Collaborating with the government can involve initiatives such as integrating eHealth platforms into oral health services [58], developing digital literacy programs to enhance eHealth literacy among the population [44], and implementing nationwide oral health campaigns leveraging digital technologies [59]. These research directions, supported by the government, can contribute to advancing eHealth strategies and improving oral health outcomes in Saudi Arabia. Further research directions in the context of eHealth and oral health in Saudi Arabia can also explore the integration of artificial intelligence (AI) in medical and dental appointments, as well as diagnostics. As highlighted in recent studies, the utilization of AI technologies has the potential to improve access to healthcare, optimize appointment scheduling, and enhance disease diagnosis [60,61].

This study has several limitations that should be acknowledged. First, the generalizability of the findings may be limited as the study was conducted at a single center, the Taibah University College of Dentistry Hospital, and the sample may not represent the broader population. Second, the cross-sectional design prevents establishing causality or determining the direction of the relationship between eHealth literacy and oral health outcomes. Third, reliance on self-reported measures introduces the potential for recall bias or social desirability bias. Despite efforts to adjust for confounding factors and address potential biases, there may be unmeasured confounders influencing the associations. One important limitation to consider, particularly in relation to the eHEALS tool, is the reliance on participants’ reading proficiency. Since the eHEALS questionnaire assesses individuals’ eHealth literacy, which involves understanding and interpreting health-related information online, it is crucial that participants possess a certain level of reading ability to provide accurate responses. This reliance on reading proficiency introduces the potential for biases, as participants with varying levels of literacy may interpret the questions in a different way or struggle to accurately report their eHealth literacy levels. Therefore, it is essential to recognize the impact of participants’ reading proficiency on the validity and reliability of the eHEALS tool and its potential influence on the study findings. These limitations emphasize the need for further research involving diverse populations, multi-center settings, longitudinal designs, and comprehensive assessments to strengthen the evidence base and enhance our understanding of the relationship between eHealth literacy and oral health. Despite these limitations, the study has notable strengths. These include the comprehensive assessment of eHealth literacy, oral health status, and oral health behaviors among a specific patient population, the adjustment for sociodemographic factors in the analyses, and the use of rigorous statistical methods to examine the associations. Additionally, the clinical observation of tooth count outcome in our study added considerable strength to our findings, as it provided an objective measure of oral health status. These strengths contribute to the validity and reliability of the findings and provide valuable insights into the relationship between eHealth literacy and oral health outcomes.

Further research directions in the field of eHealth literacy and oral health can enhance our knowledge and guide future interventions. Future studies should evaluate the effectiveness of eHealth literacy interventions, exploring different approaches and comparing their impact on oral health outcomes, as most of the studies published to date have not examined oral health [56,57]. Longitudinal research is needed to assess the long-term effects of improving eHealth literacy on sustained oral health behaviors, as most of the studies that assessed eHealth literacy and health outcomes were cross-sectional studies [40]. Furthermore, objective measures of dental hygiene such as the dental plaque index should be considered in future studies. The role of socioeconomic factors in the relationship between eHealth literacy and oral health should be investigated to address health disparities [62]. Moreover, in our study, we used eHEALS threshold values based on previous validation studies, which have demonstrated the effectiveness of these thresholds in assessing eHealth literacy [25]. However, we should highlight the importance of considering other potential approaches for modeling eHEALS scores as continuous or quintile measures in future investigations. Continuous measures would provide a more nuanced understanding of the variability in eHealth literacy levels. To our knowledge, this is the first study that assessed oral eHealth literacy and oral health outcomes, so comparative studies across diverse populations and settings can provide a broader understanding and identify contextual factors related to eHealth literacy and oral health. Additionally, the development and validation of specific measurement tools for assessing eHealth literacy in the context of oral health, such as oral health literacy tools, could be an interesting potential area for future research [13,63]. By focusing on these research areas, evidence-based interventions and policies can be developed to promote eHealth literacy and improve oral health outcomes for diverse populations.

## 5. Conclusions

In conclusion, this study sheds light on the association between eHealth literacy, oral health status, and oral health behaviors among patients visiting Taibah University College of Dentistry Hospital. The findings highlight a significant association between eHealth literacy levels and a higher mean number of teeth and a lower likelihood of not regularly brushing teeth. These results emphasize the importance of improving eHealth literacy among individuals to enhance oral health outcomes. Overall, these findings have significant public health implications, suggesting the need for targeted interventions to enhance eHealth literacy and promote oral health among the population.

## Figures and Tables

**Figure 1 healthcare-11-01804-f001:**
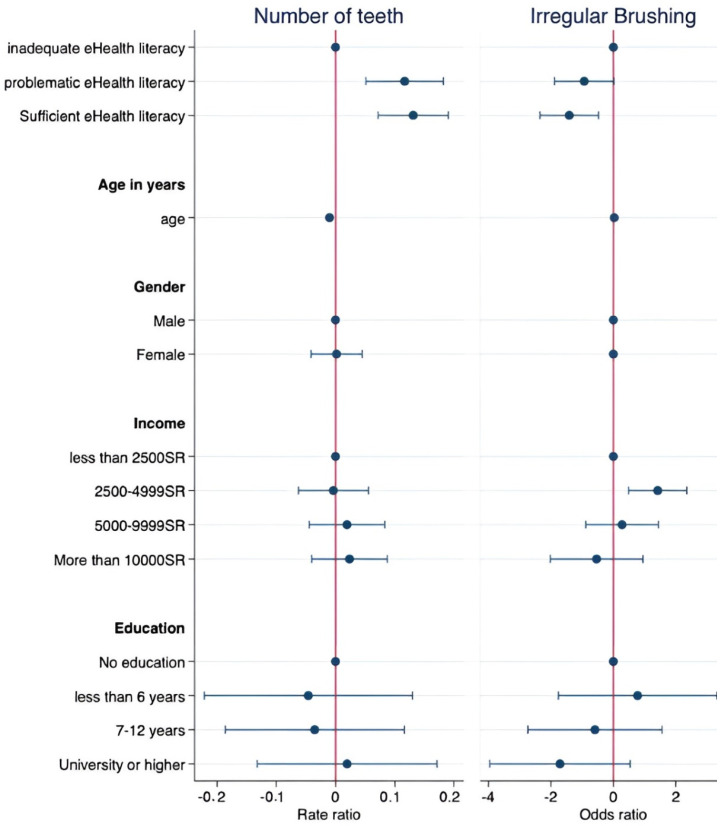
Regression coefficient plots for number of teeth and brushing frequency with respect to eHealth literacy.

**Table 1 healthcare-11-01804-t001:** Demographics and general characteristics of the study participants (n = 478).

		eHealth Literacy				
		Inadequate	Problematic	Sufficient	Total Sample	
Variable	Category	n = 81	n = 115	n = 282	n = 478	*p*-Value
Age, mean (SD)		36.8 (15.2)	33.1 (11.3)	30.6(10.8)	31.9 (12.1)	0.001
Gender, n (%)	Male	65 (80.2)	82 (71.3)	171 (60.6)	318 (66.5)	0.002
	Female	16(19.8)	33 (28.7)	111 (39.4)	160 (33.5)	
Education	No Education, n (%)	4 (4.9)	3 (2.6)	2 (0.7)	9 (1.9)	0.001
	Less than 6 years, n (%)	9 (11.1)	5 (4.3)	7 (2.5)	21 (4.4)	
	7–12 years, n (%)	35 (43.2)	50(43.5)	100 (30.5)	185 (38.7)	
	University or Higher, n (%)	33 (40.7)	57 (49.6)	173 (61.3)	263 (55.0)	
Income	Less than 2500SR, n (%)	42 (51.9)	57 (49.6)	155 (55.0)	254 (53.1)	0.623
	2500–4999SR, n (%)	13 (16.0)	24 (20.9)	37 (13.1)	74 (15.5)	
	5000–9999SR, n (%)	14 (17.3)	16 (13.9)	41 (14.5)	71 (14.9)	
	More than 10,000SR, n (%)	12 (14.8)	18 (15.7)	49 (17.4)	79 (16.5)	
Number of teeth, mean (SD)		22.6 (8.55)	26.3 (5.8)	27.6 (5.5)	26.4 (6.5)	0.001
Brushing frequency	Do not brush, n (%)	22 (27.2)	14 (12.2)	10 (3.5)	46 (9.6)	0.001
	Brush once/day, n (%)	34 (42.0)	62 (53.9)	127 (45.0)	223 (46.7)	
	Brush twice/day, n (%)	25 (30.9)	39 (33.9)	145 (51.4)	209 (43.7)	

**Table 2 healthcare-11-01804-t002:** Regression model showing the association between health literacy and number of teeth among the study sample (n = 478).

		Fully Adjusted Model		
Variable		RR	(95% CI)	*p*-Value
Age		0.99	(0.98–0.99)	0.000
Gender	Male	(Reference)		
	Female	1.00	(0.95–1.04)	0.940
Health Literacy	Inadequate	(Reference)		
	Problematic	1.12	(1.05–1.20)	0.000
	Sufficient	1.14	(1.07–1.20)	0.000
Income	Less than 25,005R	(Reference)		
	2500–49,995R	0.99	(0.93–1.05)	0.904
	5000–99,995R	1.01	(0.95–1.09)	0.554
	More than 100,005R	1.02	(0.96–1.09)	0.469
Education	No Education	(Reference)		
	Less than 6 years	0.96	(0.80–1.14)	0.608
	7–12 years	0.97	(0.83–1.12)	0.648
	University or Higher	1.01	(0.88–1.19)	0.802
RR = Rate Ratio				

**Table 3 healthcare-11-01804-t003:** Regression model showing the association between health literacy and irregular brushing among the study sample (n = 315).

		Fully Adjusted Model		
Variable		OR	(95% CI)	*p*-Value
Age		1.02	(0.99–1.06)	0.070
Gender	Male	(Reference)		
	Female	1	Empty *	Empty *
Health Literacy	Inadequate	(Reference)		
	Problematic	0.39	(0.15–1.01)	0.054
	Sufficient	0.24	(0.95–0.62)	0.003
Income	Less than 25,005R	(Reference)		
	2500–49,995R	4.14	(1.62–10.52)	0.003
	5000–99,995R	1.31	(0.41–4.22)	0.640
	More than 100,005R	0.58	(0.13–2.57)	0.477
Education	No Education	(Reference)		
	Less than 6 years	2.17	(0.17–27.57)	0.549
	7–12 years	0.55	(0.06–4.75)	0.591
	University or Higher	0.18	(0.19–1.71)	0.136
OR = Odds ratio				

* Gender is empty as there were no variations in brushing among females.

## Data Availability

Data is available upon request from authors.

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
