# Peer review of "The Association between Electronic Health Literacy and Oral Health Outcomes among Dental Patients in Saudi Arabia: A Cross-Sectional Study"

_healthcare, 2023, doi:10.3390/healthcare11121804_

Round 1

Reviewer 1 Report

The topic of the manuscript is interesting from the dental public health point of view. Still, I have some concerns about the tools used.

Title: I think it is better to add outcome after ~oral health~.

Abstract: eHealth – please explain the term for the first mentioned time and add the abbreviation.

Page 2, line 46. Please add a comma after the word  - means - and add the reference for the definition.

Line 48. Please rephrase and add the verb. For example, Mobile 48 health applications and social media.

Line 55. There is an extra space before the word ~Furthermore~

Line 168. The validated Arabic version of the Electronic Health Literacy Scale (eHEALS) – is this tool validated in the Arabic language? Please add the reference. 

Also, there are instruments for oral health literacy (e.g. REALD). Why didn`t the authors use it?

Outcome variables. Please add reference 36 after the phrase ~The number of teeth 175 was assessed through a clinical examination using the WHO method~ (line 176).

Line 176. Please add some details about the calibration process of the dentist.

Line 178. ~Remaining roots, implants and dental prostheses were not included~. I think it is confusing. The remaining root is still a natural tooth. Please remove or rephrase.

Why did the authors choose to evaluate the frequency of brushing and not assess the dental hygiene level (using a dental plaque index) during the performed clinical oral examination?

Line 213. Please add the standard deviation and the mean age range.

Line 226. Add square brackets for Table 1 in the text.

Line 257. Use ~behaviour~, there was only one assessed.

Lines 412-414. ~The findings highlight a significant association between eHealth literacy and oral health, including a higher mean number of teeth and a lower likelihood of not brushing teeth regularly~ - it is not clear, please rephrase.

Author Response

The topic of the manuscript is interesting from the dental public health point of view. Still, I have some concerns about the tools used.

 Response: Thank you for the positive feedback.

1-Title: I think it is better to add outcome after ~oral health~.

Response: Thank you for the suggestion. We have added the word outcome to the title accordingly (Page 1, line 2).

“The Association Between Electronic Health Literacy and Oral Health Outcomes Among Dental Patients in Saudi Arabia: A Cross-Sectional Study”

2-Abstract: eHealth – please explain the term for the first mentioned time and add the abbreviation.

Response: Thank you for pointing this out, we have introduced the term and added the abbreviation.  (Page 1, line 15)

 “Abstract: Objective: This cross-sectional study aimed to investigate the association between electronic health (eHealth) literacy and oral health outcomes, including the number of teeth and brushing frequency. “

3-Page 2, line 46. Please add a comma after the word  - means - and add the reference for the definition.

Response: Thank you for pointing this out, we have edited the sentence and added a reference for the definition. (Page 2, line 46)

 “Electronic health (eHealth) has been defined as healthcare services and health information provided and obtained using electronic and digital means [3] “

4-Line 48. Please rephrase and add the verb. For example, Mobile 48 health applications and social media.

Response: Thank you for pointing this out, we have rewritten the sentence.

(Page 2, line 49) “For example, implementation of various mobile health applications and social media”

5-Line 55. There is an extra space before the word ~Furthermore~

Response: Thank you for pointing this out, we have deleted the extra space.

6-Line 168. The validated Arabic version of the Electronic Health Literacy Scale (eHEALS) – is this tool validated in the Arabic language? Please add the reference. 

Response: Thank you for pointing this out, we have added the reference for the Arabic validated version of the Electronic Health Literacy Scale (eHEALS).

Wångdahl, J., Dahlberg, K., Jaensson, M. and Nilsson, U., 2021. Arabic version of the electronic health literacy scale in Arabic-speaking individuals in Sweden: prospective psychometric evaluation study. Journal of Medical Internet Research23(3), p.e24466 [24]. (Page 13, line 490)

“The validated Arabic version of the Electronic Health Literacy Scale (eHEALS) questionnaire was used to assess participants' perceived ability to find, evaluate, and use electronic health information [24]. “ (Page 3, line 134)

7-Also, there are instruments for oral health literacy (e.g. REALD). Why didn`t the authors use it?

Response: Thank you for pointing this out, we are aware of oral health literacy tools. However, as the objective of our study was to examine the association between eHealth literacy and oral health outcomes, we used eHEALS.

We appreciate your comment regarding the use of instruments for oral health literacy, such as REALD. While we acknowledge the availability of such tools, we would like to clarify our rationale for selecting the eHealth Literacy Scale (eHEALS) instead.

In our study, the primary objective was to investigate the association between eHealth literacy and oral health outcomes. As a result, we specifically focused on evaluating participants' proficiency in utilizing electronic resources for oral health information and services. Given this objective, the eHEALS has been deemed the most suitable instrument for capturing the targeted domain of eHealth literacy.

While tools like REALD assess general oral health literacy, which encompasses broader aspects of oral health knowledge and understanding, they do not specifically address the digital literacy skills required for navigating online resources. As our study aimed to explore the specific relationship between eHealth literacy and oral health outcomes in the context of digital platforms, the eHEALS provided a more direct and relevant measure for our research objectives.

8-Outcome variables. Please add reference 36 after the phrase ~The number of teeth 175 was assessed through a clinical examination using the WHO method~ (line 176).

Response: Thank you for pointing this out, we have added the reference at the end of the sentence.

 “The number of teeth and Brushing frequency. The number of teeth was assessed through a clinical examination using the WHO method [26] “ (Page 3, line 139-140)

9-Line 176. Please add some details about the calibration process of the dentist.

Response: Thank you for pointing this out, we have added more details regarding the calibration

Response: Thank you for your suggestion, we have added more details regarding the calibration

“The examiner's calibration process, following the WHO Oral Health Methods, involved two stages, ultimately yielding an excellent inter-rater reliability statistic (Kappa) of 0.98 [27,28] “(Page 3, line 143 -144)

10-Line 178. ~Remaining roots, implants and dental prostheses were not included~. I think it is confusing. The remaining root is still a natural tooth. Please remove or rephrase.

Response: Thank you for your valuable feedback regarding the exclusion of remaining roots, implants, and dental prostheses in the tooth count. We appreciate your observation and would like to provide further clarification on this matter.

In oral health epidemiology and previous studies examining tooth loss and number of teeth, it has been common practice to exclude remaining roots from the tooth count. The rationale behind this approach is that the remaining roots do not possess the full functionality of intact natural teeth. Including them in the tooth count could potentially introduce confounding factors and affect the interpretation of our results.

Peres, M.A., Antunes, J.L.F. and Watt, R.G. eds., 2021. Oral epidemiology: a textbook on oral health conditions, research topics and methods. Heidelberg: Springer.

11-Why did the authors choose to evaluate the frequency of brushing and not assess the dental hygiene level (using a dental plaque index) during the performed clinical oral examination?

Response:  Thank you for your valuable comment regarding our choice to evaluate the frequency of brushing rather than assessing dental hygiene level using a dental plaque index during the clinical oral examination. We appreciate your insightful suggestion, and we would like to provide an explanation for our approach.

In our study, the primary objective was to investigate the association between eHealth literacy and oral health outcomes, with a particular focus on individuals' health-related behaviors, including brushing frequency. While assessing dental hygiene levels using a dental plaque index would indeed provide additional information on participants' oral health status, our study design prioritized self-reported behaviors as measured through questionnaires.

We recognize that self-reported behaviors may have inherent limitations, such as recall bias or social desirability bias. However, they offer advantages in terms of ease of administration, feasibility in large-scale studies, and the ability to capture participants' perspectives and perceptions. Moreover, previous studies examining oral health behaviors have successfully utilized self-reported measures like brushing frequency to evaluate oral hygiene practices.

Nonetheless, we acknowledge the value of objective measures like dental plaque indices in assessing dental hygiene levels. In our revised manuscript, we added a sentence in the discussion to highlight the potential benefits of incorporating objective indices for dental plaque assessment in future research.

“Furthermore, objective measures of dental hygiene such as the dental plaque index should be considered in future studies.”

12-Line 213. Please add the standard deviation and the mean age range.

Response: Thank you for pointing this out, we have added the requested information in the text.

“The participants'  mean ages was 31.95 years (SD =12.1). The age range of the participants varied from 18 years to 71 years” (Page 3, line 179 -180)

13-Line 226. Add square brackets for Table 1 in the text.

Response: Thank you for pointing this out, we have added the square brackets for Table 1 in the text. (Page 3, line 196 -197)

14-Line 257. Use ~behaviour~, there was only one assessed.

Response: Thank you for pointing this out, we have used the word behaviour instead of behaviours.

“In this study, the aim was to examine the relationship between eHealth literacy and oral health status, as well as oral health behaviour, among patients who visited Taibah University College of Dentistry Hospital.”

15-Lines 412-414. ~The findings highlight a significant association between eHealth literacy and oral health, including a higher mean number of teeth and a lower likelihood of not brushing teeth regularly~ - it is not clear, please rephrase.

Response: Thank you for pointing this out, we have rewritten the sentence as follows “ The findings highlight a significant association between eHealth literacy levels and higher mean number of teeth and a lower likelihood of not brushing teeth regularly. “(Page 11, line 416 -417)

We hope that you find our responses satisfactory and that the manuscript is now acceptable for publication and we look forward to hearing from you

Reviewer 2 Report

This paper describes a cross-sectional analysis of electronic health literacy and oral health outcomes in adults in Saudi Arabia. The paper is generally well written and the subject matter is timely given oral health challenges globally. There are several issues, however, that must be addressed before this paper can be considered further. The following changes are recommended to help improve things:

·         The Background section is quite lengthy and has some areas of limited relevance to the findings. This should be shortened and focused on elements that best justify the study that was done. For example, I think much of the information on general Internet use and healthcare service trends in Saudi Arabia can be excluded.

·         There are also some relevant studies missing that would help both the Background and Discussion sections to add context to your findings, including:

o   Firmino RT, Martins CC, Faria LDS, et al. Association of oral health literacy with oral health behaviors, perception, knowledge, and dental treatment related outcomes: a systematic review and meta-analysis. J Public Health Dent. 2018;78:231-245.

o   VanWormer JJ, Tambe SR, Acharya A. Oral Health Literacy and Outcomes in Rural Wisconsin Adults. J Rural Health. 2019;35:12-21.

o   Batista MJ, Lawrence HP, Sousa MDLR. Oral health literacy and oral health outcomes in an adult population in Brazil. BMC Public Health. 2017;18:60.

·         Considerably more details are needed on the study methodology. In particular::

o   Sample – Who were the patients (e.g., patients of who/where/why?) and how did you identify them? What made this a convenience sample?

o   Recruitment – How did you identify and invite those you wanted to take the survey?

o   Exposure – How did you arrive at the eHeals threshold values for inadequate, problematic, and sufficient eHealth literacy? If these are based on prior clinical cutoffs, that should be cited. Otherwise, more sufficient justification is needed for these categories, or the authors should model eHeals scores as a continuous (or perhaps quintile) score in order to let the data drive the observed associations.   

o   Missing teeth – Most studies usually exclude ‘wisdom teeth’ in tooth counts since they are extracted for reasons other than decay (and sometimes never erupt). As such, I would suggest revising this outcome somewhat to only consider a tooth count of 0-28 teeth and exclude any third molars from consideration in this measure.

o   Tooth brushing frequency – It appears this outcome was dichotomized for analytical purposes, which is fine, but it is unclear what is included in the ‘irregular brushing’ case group. Is that no brushing, or does it also include those that brush 1/day?

·         Figure 1 adds little to the results, as it is basically a graphical representation of what is already in Tables 2 and 3. I would suggest retaining Table 3 as is, but removing Table 2. Instead, you may want to consider plotting the model-estimated number of teeth by literacy level in a Figure (as rate ratios are difficult to interpret).

I would suggest some more specificity in your strengths/limitations paragraph. For example, the tooth count outcome added considerable strength to your finings, as it was directly observed. The biggest limitation may have been self-reported literacy using the eHeals tool, which requires at least some level of reading proficiency to complete and it is unclear if this tool has been validated in an Arabic speaking population.

Author Response

This paper describes a cross-sectional analysis of electronic health literacy and oral health outcomes in adults in Saudi Arabia. The paper is generally well written and the subject matter is timely given oral health challenges globally. There are several issues, however, that must be addressed before this paper can be considered further. The following changes are recommended to help improve things:

 Response: Thank you for your positive and valuable feedback.

1-The Background section is quite lengthy and has some areas of limited relevance to the findings. This should be shortened and focused on elements that best justify the study that was done. For example, I think much of the information on general Internet use and healthcare service trends in Saudi Arabia can be excluded.

Response: Thank you for your feedback. We appreciate your suggestion to shorten and focus the Background section in our manuscript. We have carefully reviewed the section and made revisions accordingly. We have excluded information on general Internet use and healthcare service trends in Saudi Arabia that may not directly relate to the justification of our study. As a result of these edits, the word count of the Background section has been reduced from 1443 to 929, ensuring a more concise and focused presentation of the relevant background information.

2-There are also some relevant studies missing that would help both the Background and Discussion sections to add context to your findings, including:

o   Firmino RT, Martins CC, Faria LDS, et al. Association of oral health literacy with oral health behaviors, perception, knowledge, and dental treatment related outcomes: a systematic review and meta-analysis. J Public Health Dent. 2018;78:231-245.  [18]  (Page 2, line 88) (Page 8 Line 253)

o   VanWormer JJ, Tambe SR, Acharya A. O ral Health Literacy and Outcomes in Rural Wisconsin Adults. J Rural Health. 2019;35:12-21. [14]  (Page 2, line 76) (Page 8 Line 244)

o   Batista MJ, Lawrence HP, Sousa MDLR. Oral health literacy and oral health outcomes in an adult population in Brazil. BMC Public Health. 2017;18:60. [10]  (Page 2, line 72) (Page 8 Line 243)

Response: Thanks for your valuable suggestion, as suggested we have included these references in the introduction and the discussion sections.

Considerably more details are needed on the study methodology. In particular:

3-Sample – Who were the patients (e.g., patients of who/where/why?) and how did you identify them? What made this a convenience sample?

Response: Thank you for your valuable comment regarding the sample in our study. We appreciate your inquiry about the patients included and the use of convenience sampling. We have added the requested information accordingly.

“The patients included in the study were Arabic-speaking individuals aged 18 and older who sought dental treatment at the hospital and voluntarily agreed to participate. Potential participants were approached during their scheduled appointments to assess their interest and eligibility for participation. Convenience sampling was employed, where patients were selected based on their availability and willingness to participate. “(Page 3, line 119-124)

4-Recruitment – How did you identify and invite those you wanted to take the survey?

Response: Thank you for your valuable comment Potential participants were approached during their scheduled appointments to assess their interest and eligibility for participation.” (Page 3, line 121-122)

5-Exposure – How did you arrive at the eHeals threshold values for inadequate, problematic, and sufficient eHealth literacy? If these are based on prior clinical cutoffs, that should be cited. Otherwise, more sufficient justification is needed for these categories, or the authors should model eHeals scores as a continuous (or perhaps quintile) score in order to let the data drive the observed associations.   

Response: Thank you for highlighting this. In our research, we adopted the eHEALS threshold values for eHealth literacy categories based on prior studies that validated the Arabic version of the eHEALS scale. Specifically, we followed the work conducted by Wångdahl et al. (2021).

The study by Wångdahl et al. validated the Arabic version of the eHEALS scale and provided cut-off points to classify eHealth literacy levels as inadequate, problematic, and sufficient. Given the relevance of their findings to our study population, we deemed it appropriate to utilize these established cut-off values in our analysis. However, we appreciate your suggestion to consider modelling eHEALS scores as continuous or quintile scores to allow the data to drive the observed associations. In our revised manuscript, we discussed the rationale for using the eHEALS threshold values based on previous validation studies, while also acknowledging the potential benefits of exploring eHEALS scores as continuous or quintile measures in future investigations.

“Moreover, in our study, we used eHEALS threshold values based on previous validation studies, which have demonstrated the effectiveness of these thresholds in assessing eHealth literacy [25]. However, we should highlight the importance of considering other potential approaches for modelling eHEALS scores as continuous or quintile measures in future investigations. Continuous measures would provide a more nuanced understanding of the variability in eHealth literacy levels.  “ (Page 11, line 398-399)

6-   Missing teeth – Most studies usually exclude ‘wisdom teeth’ in tooth counts since they are extracted for reasons other than decay (and sometimes never erupt). As such, I would suggest revising this outcome somewhat to only consider a tooth count of 0-28 teeth and exclude any third molars from consideration in this measure.

Response: Thanks for your feedback, we appreciate your suggestion to revise the outcome by considering a tooth count of 0-28 and excluding third molars (wisdom teeth) from the analysis.

While we acknowledge the rationale behind excluding wisdom teeth from the tooth count due to their extraction for reasons other than decay and their occasional failure to erupt, it is important to note that there is variation in the inclusion/exclusion of wisdom teeth across different tooth loss studies.

Many tooth loss studies have traditionally used the full range of tooth count (0-32), including all teeth, including wisdom teeth. This approach allows for a more comprehensive assessment of overall tooth loss and considers all potential factors contributing to tooth loss, including wisdom teeth extraction.

We would like to cite a few examples of tooth loss studies that utilized the full range (0-32) in tooth count and demonstrated the association between tooth loss and various health outcomes:

  • Castrejón-Pérez, R. C., et al. (2017) examined the relationship between oral disease and the incidence of frailty in Mexican older adults using a tooth count range of 0-32.
  • Okamoto, N., et al. (2010) investigated the association between tooth loss and cognitive impairment, including mild memory impairment, in the Fujiwara-kyo study, utilizing a tooth count range of 0-32.

  • Cabrera, C., et al. (2005) explored the relationship between tooth loss and chronic disease, considering socio-economic status, in a long-term follow-up study involving women in Gothenburg, Sweden. They also used the tooth count range of 0-32.

Moreover, previous studies that have assessed Health literacy/ oral health literacy and tooth loss have also used the full range of teeth (0-32)

  • Sermsuti-Anuwat, Nithimar, and Panat Piyakhunakorn. "Association between oral health literacy and number of remaining teeth among the Thai elderly: a cross-sectional study." Clinical, Cosmetic and Investigational Dentistry(2021): 113-119.

  • Ju, Xiangqun, et al. "Effect of oral health literacy on self‐reported tooth loss: a multiple mediation analysis." Community dentistry and oral epidemiology5 (2022): 445-452

  • Geltman, Paul L., et al. "The impact of functional health literacy and acculturation on the oral health status of Somali refugees living in Massachusetts." American journal of public health8 (2013): 1516-1523.

In this study, we chose to include all teeth, including wisdom teeth, in our analysis to align with the majority of tooth loss studies and facilitate comparability across different research efforts.

7-   Tooth brushing frequency – It appears this outcome was dichotomized for analytical purposes, which is fine, but it is unclear what is included in the ‘irregular brushing’ case group. Is that no brushing, or does it also include those that brush 1/day?

Response: Thank you for your comment regarding the tooth brushing frequency outcome in our study.

For the purpose of our analysis, the category of "irregular brushing" included individuals who did not brush their teeth at all. We apologize for any confusion caused by the lack of clarity in the manuscript.

To address this concern, we revised the manuscript to explicitly state that the "irregular brushing" category refers to individuals who did not engage in any tooth brushing. (Page 4, line 166) This clarification will provide a more accurate understanding of the categorization used in our study.

We modified the text in the Data analysis part as follows:

“in the logistic regression, brushing frequency was dichotomized into “irregular brushing " category which refers to individuals who do not engage in daily tooth brushing, and "regular brushing" category includes individuals who brush their teeth at least once a day, or twice or more a day.” (Page 4, line 166 -168)

8-    Figure 1 adds little to the results, as it is basically a graphical representation of what is already in Tables 2 and 3. I would suggest retaining Table 3 as is, but removing Table 2. Instead, you may want to consider plotting the model-estimated number of teeth by literacy level in a Figure (as rate ratios are difficult to interpret).

Response: We appreciate your suggestion to remove Table 2 and consider plotting the model-estimated number of teeth by literacy level in a graphical format. However, we believe that Figure 1 provides an additional visual representation of the results, which may enhance the clarity and accessibility of the findings for readers. While Tables 2 and 3 present the numerical results, Figure 1 allows for a more intuitive understanding of the relationship between literacy level and the model-estimated number of teeth. The graphical representation can be especially helpful in identifying trends and patterns that may not be as apparent in tabular format.

9-I would suggest some more specificity in your strengths/limitations paragraph. For example, the tooth count outcome added considerable strength to your finings, as it was directly observed. The biggest limitation may have been self-reported literacy using the eHeals tool, which requires at least some level of reading proficiency to complete and it is unclear if this tool has been validated in an Arabic speaking population.

Response: Thank you for your valuable feedback, we have added some sentences in the discussion to emphasise more specificity in relation to the strengths/limitations of our study. “One important limitation to consider, particularly in relation to the eHEALS tool, is the reliance on participants' reading proficiency. Since the eHEALS questionnaire assesses individuals' eHealth literacy, which involves understanding and interpreting health-related information online, it is crucial that participants possess a certain level of reading ability to provide accurate responses. This reliance on reading proficiency introduces the potential for biases, as participants with varying levels of literacy may interpret the questions differently or struggle to accurately report their eHealth literacy levels. Therefore, it is essential to recognize the impact of participants' reading proficiency on the validity and reliability of the eHEALS tool and its potential influence on the study findings “ (Page 10, line 366 -375)

Additionally, the clinical observation of tooth count outcome in our study added considerable strength to our findings, as it provided an objective measure of oral health status “(Page 11, line 382 -384)

We hope that you find our responses satisfactory and that the manuscript is now acceptable for publication and we look forward to hearing from you

Reviewer 3 Report

Dear authors thank you for submitting the manuscript entitled “The association between electronic health literacy and oral health among dental patients: a cross-sectional study”

I have reviewed your manuscript and here is my feedback:

Since this investigation was performed in only a single country, please modify the title mentioning the country (Saudi Arabia).

In your introduction you mention the use of electronic and digital means, but you forgot to mention teledentistry, so please mention about it as well. I can see you briefly mentioned in the discussion but you can also mention in the introduction.

Currently, some pharmacies offer medical consult, medical appointments and purchasing of medicine through their phone applications, so please also mention them in your introduction.

In the outcome variables, you mentioned that a single examiner evaluated all participants (484) please mention his/her experience/expertise (example a faculty from the diagnostic clinics, a general dentist from comprehensive care clinic at the College of Dentistry, etc).

Since your study was done only in a single country, please mention further research directions with the government help in your country.

You are mentioning novel technologies so please mention if there is any artificial intelligence (AI) for medical/dental appointments and/or diagnostics provided. Here are some references that may help you:

-       Samorani M, Blount LG. Machine Learning and Medical Appointment Scheduling: Creating and Perpetuating Inequalities in Access to Health Care. Am J Public Health. 2020 Apr;110(4):440-441. doi: 10.2105/AJPH.2020.305570. PMID: 32159974; PMCID: PMC7067080.

-       Kumar Y, Koul A, Singla R, Ijaz MF. Artificial intelligence in disease diagnosis: a systematic literature review, synthesizing framework and future research agenda. J Ambient Intell Humaniz Comput. 2022 Jan 13:1-28. doi: 10.1007/s12652-021-03612-z. Epub ahead of print. PMID: 35039756; PMCID: PMC8754556

Even though the results were expected and similar studies have already shown that the lower oral health literacy the more problems are present in the patients, I believe this manuscript is important in order to confirm previous findings and to provide specific data for Saudi Arabia.

Please read the following article and see if it can help you to improve your manuscript

-       Wehmeyer MM, Corwin CL, Guthmiller JM, Lee JY. The impact of oral health literacy on periodontal health status. J Public Health Dent. 2014 Winter;74(1):80-7. doi: 10.1111/j.1752-7325.2012.00375.x. Epub 2012 Nov 2. PMID: 23121152; PMCID: PMC3800213.

English language was fine

Author Response

Dear authors thank you for submitting the manuscript entitled “The association between electronic health literacy and oral health among dental patients: a cross-sectional study”

I have reviewed your manuscript and here is my feedback:

Response: Thank you for your positive feedback

1-Since this investigation was performed in only a single country, please modify the title mentioning the country (Saudi Arabia).

 Response: Thank you for your suggestion. We agree and we have modified the title accordingly.

“ The Association Between Electronic Health Literacy and Oral Health Outcomes Among Dental Patients in Saudi Arabia: A Cross-Sectional Study” (Page 1, line 2 -3)

2-In your introduction you mention the use of electronic and digital means, but you forgot to mention teledentistry, so please mention about it as well. I can see you briefly mentioned in the discussion but you can also mention in the introduction.

 Response: Thank you for your suggestion. We have added a paragraph regarding Teledentistry in the Introduction.

“Indeed, the term Teledentistry is considered an example of such a concept that refers to the provision of dental care, advice, or treatment using online communication technology, such as video conferencing, remote monitoring, and online consultations [21]. This innovation has revolutionized the dental industry, improving access to oral care, and promoting better oral health behavior. In fact, Teledentistry plays a crucial role in promoting oral Health literacy, as patients engage in virtual consultations and access oral health information online, they become adept at navigating digital health resources and making informed decisions about their oral care [21]. “(Page 3, line 96 -103)

3-Currently, some pharmacies offer medical consult, medical appointments and purchasing of medicine through their phone applications, so please also mention them in your introduction.

 Response: Thank you for your suggestion. We have included these application in the paragraph that addresses Teledentistry in the Introduction.

‘“Indeed, the term Teledentistry is considered as an example of such concept that refers to the provision of dental care, advice, or treatment using online communication technology, such as video conferencing, remote monitoring, and online consultations [21]” (Page 3, line 96 -98)

4-In the outcome variables, you mentioned that a single examiner evaluated all participants (484) please mention his/her experience/expertise (example a faculty from the diagnostic clinics, a general dentist from comprehensive care clinic at the College of Dentistry, etc).

 Response: Thank you for your suggestion. We have added the expertise of the examiner in the text.

“(dental intern at the College of Dentistry, Taibah University)” (Page 3, line 141)

5-Since your study was done only in a single country, please mention further research directions with the government help in your country.

 Response: Thanks for your feedback, we have added the following paragraph to the discussion:

“Considering that our study focused on eHealth and oral health in a single country, Saudi Arabia, there are several key directions for further research with the support of the government. Collaborating with the government can involve initiatives such as integrating eHealth platforms into oral health services[58], developing digital literacy programs to enhance eHealth literacy among the population [44], and implementing nationwide oral health campaigns leveraging digital technologies [59]. These research directions, supported by the government, can contribute to advancing eHealth strategies and improving oral health outcomes in Saudi Arabia.” (Page 10, line 345-352)

6-You are mentioning novel technologies so please mention if there is any artificial intelligence (AI) for medical/dental appointments and/or diagnostics provided. Here are some references that may help you:

  • Samorani M, Blount LG. Machine Learning and Medical Appointment Scheduling: Creating and Perpetuating Inequalities in Access to Health Care. Am J Public Health. 2020 Apr;110(4):440-441. doi: 10.2105/AJPH.2020.305570. PMID: 32159974; PMCID: PMC7067080.
  • Kumar Y, Koul A, Singla R, Ijaz MF. Artificial intelligence in disease diagnosis: a systematic literature review, synthesizing framework and future research agenda. J Ambient Intell Humaniz Comput. 2022 Jan 13:1-28. doi: 10.1007/s12652-021-03612-z. Epub ahead of print. PMID: 35039756; PMCID: PMC8754556

Response: Thank you for highlighting this, we added a paragraph highlighting this and we cited the suggested references:

Further research directions in the context of eHealth and oral health in Saudi Arabia can also explore the integration of artificial intelligence (AI) in medical and dental appointments as well as diagnostics. As highlighted in recent studies, the utilization of AI technologies has the potential to improve access to healthcare, optimize appointment scheduling, and enhance disease diagnosis [60,61].” (Page 10, line 352 -356)

7-Even though the results were expected and similar studies have already shown that the lower oral health literacy the more problems are present in the patients, I believe this manuscript is important in order to confirm previous findings and to provide specific data for Saudi Arabia.

Response: Thank you for your positive feedback, we appreciate your recognition of the importance of this manuscript in confirming previous findings and providing specific data for Saudi Arabia

Please read the following article and see if it can help you to improve your manuscript

  • Wehmeyer MM, Corwin CL, Guthmiller JM, Lee JY. The impact of oral health literacy on periodontal health status. J Public Health Dent. 2014 Winter;74(1):80-7. doi: 10.1111/j.1752-7325.2012.00375.x. Epub 2012 Nov 2. PMID: 23121152; PMCID: PMC3800213.

Response: Thank you for your suggestion, we have cited the article in the discussion section.

“When compared to individuals with greater literacy, those with poorer health literacy and oral health literacy brush their teeth less frequently compared to those with higher literacy [10,28-32]. They were also shown to have more tooth loss over time [30-32] and worse periodontal disease status [33]” (Page 8, line 240-242)

We hope that you find our responses satisfactory and that the manuscript is now acceptable for publication and we look forward to hearing from you

Round 2

Reviewer 1 Report

The authors addressed all the comments.

Reviewer 2 Report

Responses and updates are acceptable. Thanks.